# Identifying Corn Lodging in the Mature Period Using Chinese GF-1 PMS Images

**Xianda Huang** [1,2], **Fu Xuan** [1,2], **Yi Dong** [1,2], **Wei Su** [1,2,*], **Xinsheng Wang** [1,2], **Jianxi Huang** [1,2], **Xuecao Li** [1,2], **Yelu Zeng** [1,2], **Shuangxi Miao** [1,2] **and Jiayu Li** [1,2]

1   College of Land Science and Technology, China Agricultural University, Beijing 100083, China
2   Key Laboratory of Remote Sensing for Agri-Hazards, Ministry of Agriculture, Beijing 100083, China
*   Correspondence: suwei@cau.edu.cn

**Abstract:** Efficient, fast, and accurate crop lodging monitoring is urgent for farmers, agronomists, insurance loss adjusters, and policymakers. This study aims to explore the potential of Chinese GF-1 PMS high-spatial-resolution images for corn lodging monitoring and to find a robust and efficient way to identify corn lodging accurately and efficiently. Three groups of image features and five machine-learning approaches are used for classifying non-lodged, moderately lodged, and severely lodged areas. Our results reveal that (1) the combination of spectral bands, optimized vegetation indexes, and texture features classify corn lodging with an overall accuracy of 93.81% and a Kappa coefficient of 0.91. (2) The random forest is an efficient, robust, and easy classifier to identify corn lodging with the F1-score of 0.95, 0.92, and 0.95 for non-lodged, moderately lodged, and severely lodged areas, respectively. (3) The GF-1 PMS image has great potential for identifying corn lodging on a regional scale.

**Keywords:** corn lodging; GF-1 PMS image; vegetation index; texture feature; random forest

## 1. Introduction

Lodging is a major disaster for the crop, resulting in yield reduction and quality degradation [1,2]. The strong winds or heavy rain/hail will lead to different lodging, including stem lodging, which is the bending of the crop stem from their upright position, and root lodging, which is the failure of the root–soil anchorage system [3,4]. Concerning stem lodging, the corn plants may be recovered by their self-recovery ability. The yield reduction and quality degradation are inevitable for severe or root lodging, especially in the mature period. So, the farmers, agronomists, insurance loss adjusters, and policymakers need fast and accurate information on the location and severity of crop lodging on a regional scale.

The traditional method of acquiring information on the location and severity of crop lodging relies on the measurement in field campaign, which is time-consuming, laborious, and subjective and cannot be performed on a regional scale. Fortunately, remote sensing provides a timely and reliable method, including high-spatial-resolution and multi-spectral features, for acquiring crop lodging information across large areas [4–6]. The earliest study can be traced back to the identification of winter wheat lodging using microcomputer-assisted video images [7]. Subsequently, there are many studies using ground-based, space-borne, and airborne remote sensing images to monitor crop lodging. The studies monitoring crop lodging using ground-based images provide a valuable identification of the behavior of the remote sensing signal on crop lodging using small-scale experiments [8–10]. With the development of remote sensing and unmanned drone techniques, unmanned vehicle (UAV) images are used are identifying crop lodging with high spatial resolution on a farm or many field plots [11–15]. The space-borne satellite images have superiority in monitoring crop lodging on a regional scale. Two satellite images are used for crop

lodging monitoring, including active SAR and passive optical images. Yang et al. [16], Chen et al. [17], and Zhao et al. [18] explore Radarsat-2 quad-polarimetric images to monitor the lodging of wheat and sugarcane. Chauhan et al. [2] classify the wheat lodging severity using Radarsat-2 and Sentinel-1 images, and they found that the SAR-based metrics can capture the crop lodging severity on a regional scale. Furthermore, there is an obvious difference between crop lodging and non-lodging areas based on spectral reflectance and its derived vegetation indexes and texture metrics [19–21]. Guan et al. [22] explore the potential of aggregating Sentinel-2 metrics, including selected spectral bands and vegetation indexes with a spatial resolution of 10 m. Referenced by these works, we want to know if the high-spatial-resolution optical image with a meter-level resolution has greater potential for crop lodging monitoring. Therefore, in this study, we explore the potential of Chinese GF-1 PMS images for crop lodging identification.

Gaofen1 (GF-1, launched on 26 April 2013) is the first satellite of the China High-resolution Earth Observation System (CHEOS) project. One major application of GF-1 images is in agricultural monitoring. Two sensors are carried on the GF-1 satellites, including wide field view (WFV) and panchromatic multispectral sensors (PMSs) cameras [23,24]. Zhou et al. [25], Chen et al. [26], and Qu et al. [27] used the GF-1 WFV image with 16 m spatial resolution to monitor crop lodging. The panchromatic band of the GF-1 PMS has a spatial resolution of 2 m. Therefore, the GF-1 PMS image is used to explore the potential of a satellite image with a meter-level resolution for crop lodging monitoring in this study. There is severe corn lodging in the middle of September when the mature period was hit by three consecutive typhoons, including *Bawei*, *Mesak*, and *Poseidon*, in Lishu County, Jilin Province, China. Therefore, corn lodging identification in Lishu County using GF-1 PMS images is performed in this study. The objectives are as follows:

(1)  Exploring the potential of texture features calculated from GF-1 PMS images for corn lodging classification, including non-lodged, moderately lodged, and severely lodged areas.
(2)  Finding the optimized spectral bands, vegetation indexes, and textural features to improve corn lodging classification accuracy.
(3)  Improving the efficiency and robustness of corn lodging classification based on GF-1 PMS images using an optimized machine learning approach.

## 2. Study Area and Data Sources

### 2.1. Study Area and In Situ Measurements

2.1.1. Study Area

The study area is located in Lishu County, Jilin Province, China, ranging from 123°45′E, 43°02′N to 124°53′E, 43°46′N, covering 4209 km$^2$ (Figure 1). The annual average precipitation is 614 mm, and the annual average temperature is 6.9 °C belonging to the continental monsoon climate zone with distinct seasonal changes. The cropland area is 2649 km$^2$, and 90% is the corn planted area. The growing season of corn ranges from the end of April to the end of September. Black soil dominates the whole study area, and conservation tillage is performed to protect the black soil. Jilin Province was hit by three typhoons, including *Bawei*, *Mesak*, and *Poseidon*, from August to early September 2020. Thus, there was lodging of the corn plants in Lishu County.

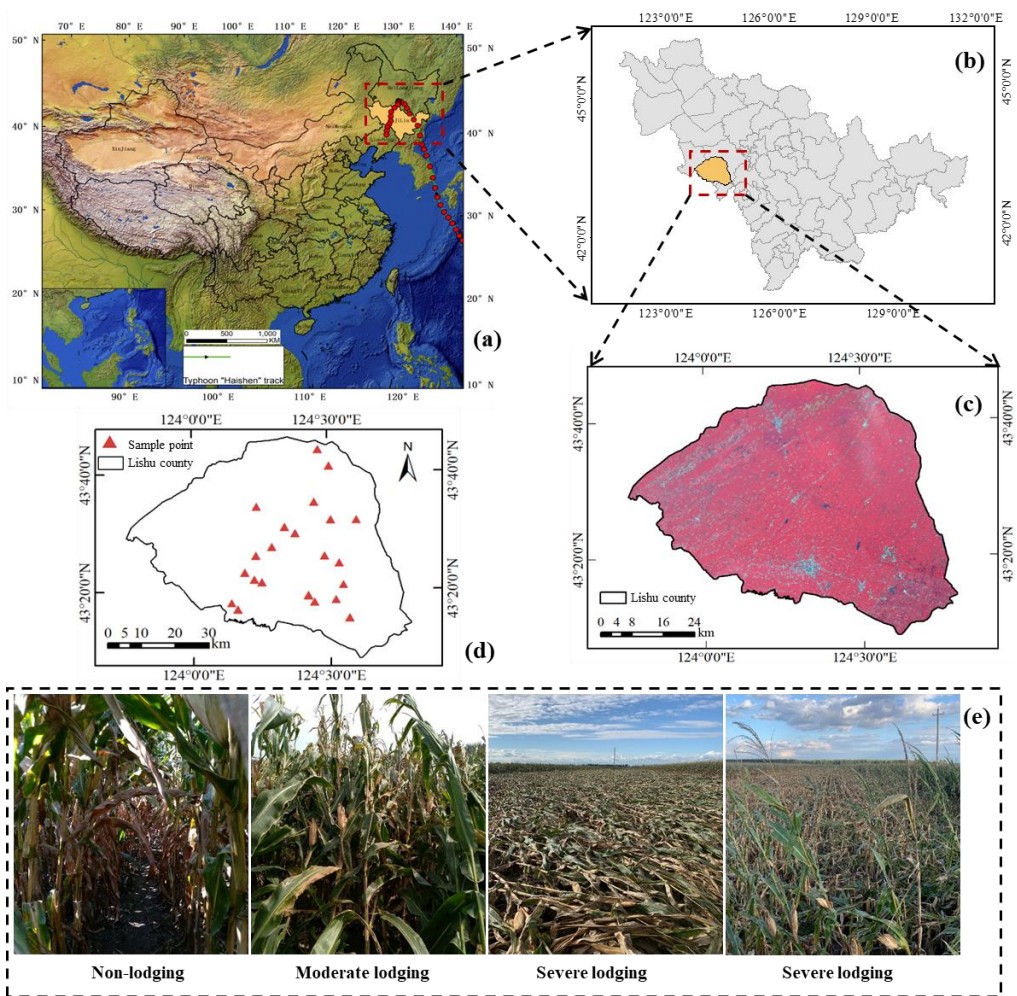

**Figure 1.** The study area (**a**,**b**), mosaic GF-1 PMS image (**c**) acquired before the typhoon (R: Band1, G: Band2, B: Band3), location of measured plots (**d**), and photographs (**e**) taken during a field campaign.

### 2.1.2. In Situ Measurements

For modeling and validating the lodging area, we measured the lodged area using a Huace LT700H real-time kinematic (RTK) GPS receiver (Huace Ltd., Shanghai, China) and DJI Inspire2 UAV with (R, G, B) bands from the 11th to 17th September 2020. Every lodged corn planted area was located using a Huace LT700H RTK GPS receiver. The inclinometer is used to measure the corn lodging angle. The measured plots are shown in Figure 1d with photographs taken during the field campaign (Figure 1e). There are 37 plots with severe lodging, 37 plots with moderate lodging, and 14 plots with non-lodging. The lodging angle is used to classify the lodging severity as severe lodging ($\alpha \geq 60°$), moderate lodging ($30° < \alpha < 60°$), and non-lodging, where $\alpha$ is the measured lodging angle using an inclinometer.

### 2.1.3. UAV Collection

Because the lodged area cannot be reached by walking, we used the DJI Inspire2 drone to collect the high spatial UAV images for the visual interpretation of samples in conjunction with the ground survey points. A ZENMUSE X5S camera is carried on the DJI Inspire2 drone for collecting high spatial visible images, including blue, green, and red bands. The GJI GO 4 software on an iPad platform is used to set the flight route and flight parameters for corn lodging identification before shooting. The acquisition parameters for UAV image collection are presented in Table 1.

**Table 1.** Acquisition parameters for UAV image collection.

| Parameters | Value |
|---|---|
| Flight height/m | 50 |
| Speed of flight/(m · s$^{-1}$) | 5 |
| Before-and-after overlap/% | 80 |
| Side overlap/% | 65 |
| Spatial resolution/(cm · px$^{-1}$) | 1.4 |

Many pictures are acquired after the drone flight is finished, and the advanced agriculture mapping software Pix4Dfields for aerial crop analysis and digital farming is used to perform image mosaic and geographical registration. Examples of the processed UAV images are shown in Figure 2: the light fields are lodged corn-planted areas, and the varied green fields are the non-lodged areas. The field in Figure 2a is oriented north–south, and the field in Figure 2a is oriented east–west. We can see that the textural characteristics and textural orientations of the lodged areas differ for the different fields in Figure 2a,b.

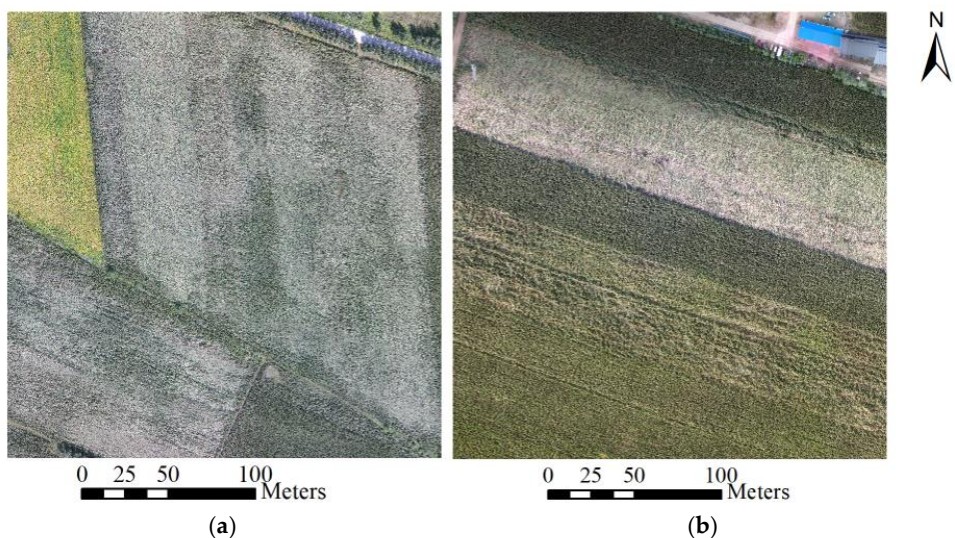

**Figure 2.** UAV images collected during the field campaign. The field of North-south (**a**) and east-west (**b**) orientations.

### 2.2. GF-1 PMS Images and Pre-Processing

The high-spatial-resolution GF-1 PMS images are used to identify corn lodging areas. Image preprocessing for radiometric calibration, atmospheric correction, orthorectification, fusion, and stitching is performed before lodging area classification. The characteristics of the GF-1 PMS images used for corn lodging identification in this study are shown in Table 2. In this study, we use four machine-learning methods, including support vector machine, random forest, naive Bayesian, and BP neural network, to identify lodged corn-planted areas using the GF-1 PMS images. There should be adequate samples for machine learning methods. Therefore, we collect three samples using the visual interpretation method combining ground-truth points and UAV imagery data: non-lodging, moderate lodging, and severe lodging areas. There are a total of 4526 samples, and each kind of sample is randomly divided into training samples (70%) and validation samples (30%). The number of these three kinds of samples used for training and validation is presented in Table 3.

**Table 2.** Characteristics of GF-1 PMS images used for corn lodging identification.

| No. | Name | Acquisition Date |
|---|---|---|
| 1 | GF1_PMS1_E124.0_N43.9_20200926_L1A0005087667 | |
| 2 | GF1_PMS1_E124.0_N43.6_20200926_L1A0005087679 | |
| 3 | GF1_PMS1_E123.9_N43.3_20200926_L1A0005087685 | 26 September 2020 |
| 4 | GF1_PMS2_E124.5_N43.8_20200926_L1A0005087811 | |
| 5 | GF1_PMS2_E124.4_N43.5_20200926_L1A0005087820 | |
| 6 | GF1_PMS2_E124.3_N43.2_20200926_L1A0005087822 | |

**Table 3.** Number of training and validation samples used for machine learning.

| Code | Type | Training Samples/Pixel | Validation Samples/Pixel | Total/Pixel |
|---|---|---|---|---|
| 0 | Non-lodging | 972 | 433 | 1405 |
| 1 | Moderate lodging | 1152 | 501 | 1653 |
| 2 | Severe lodging | 1045 | 423 | 1468 |

## 3. Methodology

The flowchart for identifying corn lodging in the mature period using GF-1 PMS images is illustrated in Figure 3. We collected samples using UAV imagery and measured data from field, and the samples served as training and validation data for the model. After the pre-processing of GF-1 PMS images, three kinds of image features including spectral bands, vegetation indexes, and textural features are built. All these features are ranked by importance and the sensitive features are optimized and combined. There are four groups of optimized features that will be input into five machine learning classifiers including the support vector machine (SVM), random forest (RF), naive Bayesian (NB), BP neural network, and Extreme Gradient Boosting (XGboost). Lastly, the optimized model will be used for corn lodging identification and mapping.

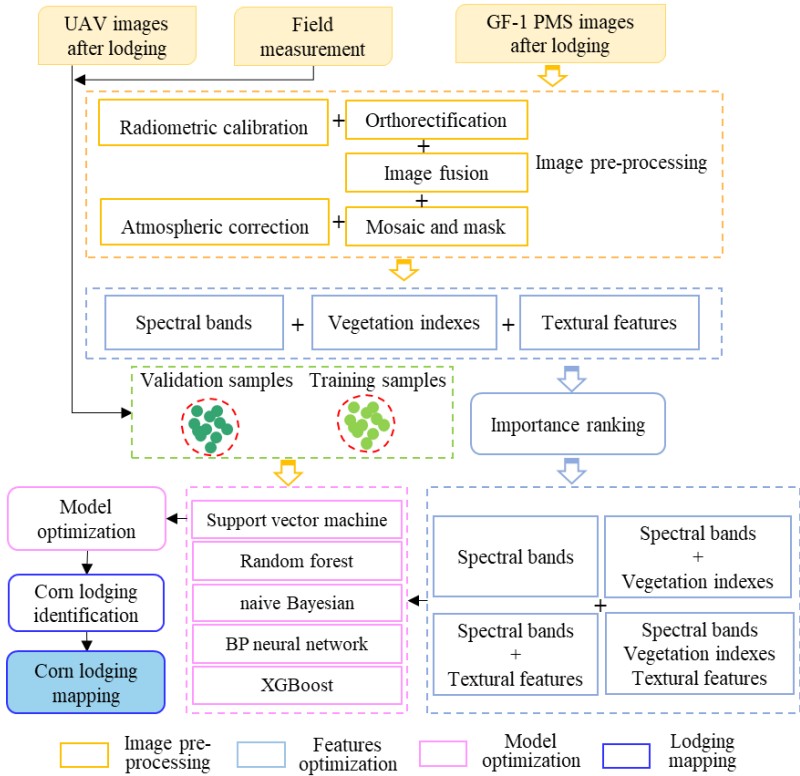

**Figure 3.** Schematic diagram of corn lodging identification using GF-1 PMS images.

### 3.1. Image Features

Three image features are used to identify corn-lodged areas, including spectral bands, vegetation indexes, and textural features. Their definiens and applications for identifying corn lodged area using GF-1 PMS high-spatial-resolution images are as follows.

### 3.1.1. Spectral Features

The four spectral bands, including visible and NIR bands of the GF-1 PMS images, are used for identifying lodged and non-lodged areas in this study. There are obvious color and tone differences between lodged and non-lodged areas, as shown in Table 4. The visual interpreting flags in Table 4 show that the color of the lodged area is brighter than the non-lodged area. This may result from the brightness of the back side of corn leaves. Comparably, the non-lodged corn planted area is dark green or dark red, the moderately lodged area is bright green or bright red, and the severely lodged area is brighter green or red. These color and tone differences are used for identifying lodged and non-lodged areas.

**Table 4.** Spectral features of non-lodging, moderate lodging, and severe lodging from GF-1 PMS images.

| Code | Type | True Color Image | False Color Image |
|---|---|---|---|
| 1-4 1 | Non-lodging | | |
| 1-4 2 | Moderate lodging | | |
| 1-4 3 | Severe lodging | | |

### 3.1.2. Vegetation Indexes

The vegetation index can quantify the growing condition and vegetation coverage, improving the identification of non-lodged, moderately lodged, and severely lodged areas [28]. There are 13 vegetation indexes used to identify lodged and non-lodged areas in this study. The expression and reference of these 13 vegetation indexes are presented in Table 5.

There will be redundancy within these 13 vegetation indexes, and the redundancy will result in big computing and does not help to improve the accuracy of identification. So, we analyze the importance of these vegetation indexes using the Gini index before the lodged area identification. The vegetation indexes with major importance are used, and the vegetation indexes with minor importance will be removed during the lodged area identification. The Gini index is calculated as follows:

$$Gini = 1 - \sum_{k=1}^{K} p_k^2 \tag{1}$$

where $K$ is the number of classes and $p_k$ is the probability of class $k$. The Gini value ranges from 0 to 1, which can be used to quantify the importance of image features. A larger Gini coefficient suggests an important image feature, and a smaller Gini coefficient indicates that the analyzed feature is less different from other features and less important. The importance-ranked results of the 13 vegetation indexes are presented in Section 4.2.

**Table 5.** Vegetation indexes for corn lodging identification.

| Abbreviations | Full Names | Expressions | References |
|---|---|---|---|
| NDVI | Normalized Difference Vegetation Index | $NDVI = \frac{\rho_{rir}-\rho_{red}}{\rho_{rir}+\rho_{red}}$ | [29] |
| EVI | Enhanced Vegetation Index | $EVI = \frac{2.5\times(\rho_{nir}-\rho_{red})}{\rho_{nir}+6\times\rho_{red}-7.5\times\rho_{blue}+1}$ | [30] |
| RVI | Ratio Vegetation Index | $RVI = \frac{\rho_{nir}}{\rho_{red}}$ | [29] |
| DVI | Difference Vegetation Index | $DVI = \rho_{nir}-\rho_{red}$ | [29] |
| TVI | Triangular Vegetation Index | $TVI = 60\times(\rho_{nir}-\rho_{green})-100\times(\rho_{red}-\rho_{green})$ | [4] |
| ARVI | Atmospheric Resistant Vegetation Index | $ARVI = \frac{\rho_{nir}-2\times\rho_{red}+\rho_{blue}}{\rho_{nir}+2\times\rho_{red}-\rho_{blue}}$ | [29] |
| GNDVI | Green Normalized Difference Vegetation | $GNDVI = \frac{\rho_{nir}-\rho_{green}}{\rho_{nir}+\rho_{green}}$ | [4] |
| GRVI | Green Ratio Vegetation Index | $GRVI = \frac{\rho_{nir}}{\rho_{green}}-1$ | [4] |
| VDVI | Visible-Band Difference Vegetation Index | $VDVI = \frac{2\times\rho_{green}-\rho_{red}-\rho_{blue}}{2\times\rho_{green}+\rho_{red}+\rho_{blue}}$ | [4] |
| SAVI | Soil Adjusted Vegetation Index | $SAVI = \frac{(\rho_{nir}-\rho_{red})\times(1+L)}{\rho_{nir}+\rho_{red}+L}$ | [29] |
| NLI | Nonlinear Vegetation Index | $NLI = \frac{\rho_{rir}^2-\rho_{red}}{\rho_{nir}^2+\rho_{red}}$ | [31] |
| RDVI | Renormalized Difference Vegetation Index | $RDVI = \frac{\rho_{nir}-\rho_{red}}{\sqrt{\rho_{nir}+\rho_{red}}}$ | [4] |
| SIPI | Structure Insensitive Pigment Index | $SIPI = \frac{\rho_{nir}-\rho_{blue}}{\rho_{nir}+\rho_{red}}$ | [4] |

In order to determine the optimal combination of vegetation indices, according to the importance ranking results of the vegetation indices, the features are combined one by one to construct a model, and the OOB error of the random forest model for each combination is obtained [32], the number of image features is determined by comparing the OOB errors of each combination. We determine the optimal vegetation index combination through the importance ranking results and the OOB error.

### 3.1.3. Textural Features

Figures 2 and 3 show the prominent textural characteristics within the lodged area, and the textural orientation corresponds to the lodging direction of corn plants. Therefore, we explore the protentional textural features of the GF-1 PMS images to identify the lodged corn planted area. Referencing our previous work about textural features for land cover classification [33], the GLCM approach is used to produce textural features for identifying lodged and non-lodged areas in this study. The GLCM approach was developed by Haralick [34], which can fully take into account the spectral and spatial pattern of image grey values [35,36]. There are eight popular GLCM textural features, including Mean, variance (VAR), correlation (COR), contrast (CON), dissimilarity (DIS), homogeneity (HOM), angular second moment (ASM), and entropy (ENT) [30]; these features were used

in this study. When selecting optimal texture features for classifying corn lodging, the Jeffries–Matusita (J.M.) distance indices are used to measure the separability of non-lodged, moderately lodged, and severely lodged areas from the texture features. The J.M. distance is calculated as follows:

$$JM_{ij} = 2\sqrt{1 - e^{-B}} \tag{2}$$

$$JM_{ij}B = \frac{1}{8}(m_i - m_j)^{\mathrm{T}}\left(\frac{C_i + C_j}{2}\right)^{-1}(m_i - m_j) + \frac{1}{2}\ln\frac{\left|\frac{C_i + C_j}{2}\right|}{\sqrt{|C_i| \times |C_j|}} \tag{3}$$

where, $B$ is the Bhattacharyya distance between class $i$ and class $j$; $m_i$ and $m_j$ are the mean distances between class $i$ and class $j$, respectively; and $C_i$ and $C_j$ are the covariance matrixes of class $i$ and class $j$, respectively.

### 3.2. Machine Learning Algorithms

Five machine learning methods are used to identify the lodged area in this study, including support vector machine (SVM), random forest (RF), naive Bayesian (NB), BP neural network, and Extreme Gradient Boosting (XGBoost).

The SVM is a popular algorithm for solving the problem of pattern recognition with clear connections to the underlying statistical learning theory [37]. The classification results of the SVM model are affected by many parameters, and the two most important parameters are the error penalty parameter C and the kernel function [38]. The choice of the kernel is vital for the pattern recognition accuracy using SVM, which links the problems they are designed for with a large body of existing work on kernel-based methods. Three kernel functions are used to identify the lodged area, including the polynomial kernel function (poly), RBF kernel function (rbf), and Sigmoid kernel function (sigmoid).

A random forest is an ensemble of trees that lets them vote for the most popular class, bringing significant improvements in classification accuracy [39]. So, the random forest approach is used for identifying the lodged area. For the random forest method, forests are a combination of tree predictors. Each tree depends on the values of a random vector sampled independently with the same distribution for all trees in the forest [40]. During the lodged area identification, two vital parameters should be optimized: the number of decision trees and the maximum number of features used for generating each decision tree. The number of decision trees is decided from 1 to the number of training samples. There are 3169 training samples in this study. Therefore, the range of the number of decision trees is set as (13,200) in this study. The maximum number of image features is decided by comparing classification accuracy using 50, 100, 500, 1000, and 2000 decision trees.

The Naive Bayesian classifier is based on Bayesian theory [41], which assumes that the predicted variables are Gaussian-distributed and all are independent. For the lodged area identification, there are $n$-dimensional features for each sample of a lodged area which can be set as $X = \{x_1, x_2, \cdots, x_n\}$. There are three kinds of classes set as $C_i (1 \leq i \leq 3)$ for the non-lodged areas, moderately lodged areas, and severely lodged areas. The analyzed sample should be predicted as the class with the highest posterior probability. The lodged area classification can be calculated as:

$$P(C_i|X) = \frac{P(C_i)\prod_{k=1}^{n}P(x_k \mid C_i)}{P(X)} \tag{4}$$

where $P(x_k \mid C_i)$ is calculated using the training samples of non-lodged, moderately, and severely lodged areas. $X$ will be predicted to be $C_i$ only if $P(C_i \mid X) > P(C_j \mid X), 1 \leq j \leq 3, j \neq i$. The advantage of the Naive Bayes classifier is that no super parameters should be set, which is easy to complete.

The BP neural network is a multilayer feedforward neural network that corrects network parameters by backpropagating produced errors. The BP neural network has three layers: input, hidden, and output. In addition to determining the number of layers, the number of neural nodes in each layer is a vital parameter. For the corn lodged area

classification, the number of nodes in the input layer is determined by the number of features in the feature set of the non-lodged area, moderately lodged area, and severely lodged area. The number of output layer nodes is set to 3. Therefore, only the number of nodes in the hidden layer should be determined, which is set using the following equation:

$$s = \sqrt{0.43mn + 0.12n^2 + 2.54m + 0.77n + 0.35} + 0.51 \tag{5}$$

where $s$ is the number of nodes in the hidden layer, $m$ is the number of nodes in the input layer, and $n$ is the number of nodes in the output layer. The number of nodes in the input layer, hidden layer, and output layer for the combination of spectral features, spectral features + vegetation indexes, spectral + textural features, and spectral + textural features + vegetation indexes are presented in Table 6. Within the training process of BP neural networks, the training cycle is set at 500, the optimizer is set at Adam, the learning rate is set at 0.001, and the batch size is set at 100.

**Table 6.** The number of nodes for four kinds of feature combinations.

| Feature Combinations | Nodes in the Input Layer | Nodes in Hidden Layer | Nodes in the Output Layer |
|---|---|---|---|
| spectral features | 4 | 5 | 3 |
| spectral features + vegetation indexes | 13 | 8 | 3 |
| spectral + textural features | 9 | 7 | 3 |
| spectral + textural features + vegetation indexes | 18 | 9 | 3 |

XGBoost is a novel implementation of gradient-boosted decision trees [42]. It is based on augmented ensemble techniques that combine a set of weak learners to develop a strong learner with an additive strategy. XGBoost takes the direction of the gradient descent of the loss function as the optimization goal, and the new learner is built on the direction of the gradient descent of the loss function of the previous learner. The main parameters determining the model structure include the number of gradient-boosted trees and the maximum tree depth [43]. The optional parameters for the number of gradient-boosted trees are 50, 100, 150, and 200, respectively. The maximum tree depth options are 5, 7, 9, 12, 15, 17, and 25. Parameter selection will be determined using a grid search method based on cross-validation.

The confusion matrix validates the classification results of non-lodged, moderately, and severely lodged areas, and the overall accuracy (OA), Kappa coefficient (k), and F1 score are used to assess the classification accuracy.

## 4. Results and Analysis

### 4.1. Spectral Difference Analysis

To clearly show the spectral difference between the lodged and non-lodged areas quantitatively, the boxplot and spectral difference of the non-lodged, moderately lodged, and severely lodged areas within visible and NIR bands are presented in Figure 4. From Figure 4a, we can see that there is an obvious reflectance increase and a broadened standard deviation range for the lodged area in all four spectral bands including the blue, green, red, and NIR bands compared with the reflectance of the non-lodged area. The reflectance increase in the NIR band is the highest. Figure 4b shows that the reflectance of the blue, green, red, and NIR bands increases from the non-lodged areas to the moderately lodged and severely lodged areas. This result is consistent with the findings for the UAV image labels in Figure 3 and the spectral features on the GF-1 PMS images in Table 4.

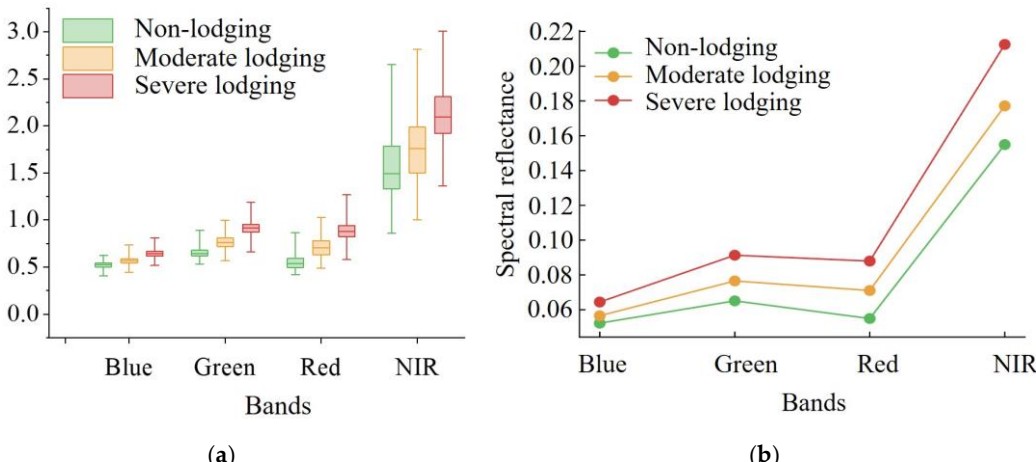

(**a**)   (**b**)

**Figure 4.** The boxplot (**a**) and spectral difference (**b**) of the non-lodged, moderately lodged, and severely lodged areas within four spectral bands.

### 4.2. Vegetation Index Difference Analysis

The *Gini* index is calculated to rank the 13 vegetation indexes in Table 5. The ranking results in Figure 5a shows that the feature with the highest importance is ARVI. In addition, the four features of DVI, NDVI, RVI, and TVI were ranked high in importance, and the remaining eight features (NLI, SIPI, VDVI, GNDVI, SAVI, EVI, GRVI, and RDVI) have relatively low scores. In order to determine the optimal combination of the vegetation indexes, the image features are combined one by one according to the ranking results of importance. The out-of-bag (OOB) error of the random forest classifier is used to determine the number of image features. Figure 5b is the line chart of the number of vegetation indexes with the OOB error. Firstly, the OOB error decreases quickly from 0.516 to 0.177 after the first important ARVI feature is joined. Secondly, the OOB error decreases slower from 0.177 to 0.115 after the second important DVI feature is joined. Next, the OOB error decreases from 0.115 to 0.113 with a decreased value of only 0.002. Subsequently, the OOB error decreases slowly with the joining of more vegetation indexes. The OOB error reaches a minimum of 0.069 when the number of vegetation indexes reaches nine. After that, the OOB error is stabilized despite more vegetation indexes being joined. Therefore, the optimized vegetation index combination is ARVI, DVI, NDVI, RVI, TVI, NLI, SIPI, VDVI, and GNDVI.

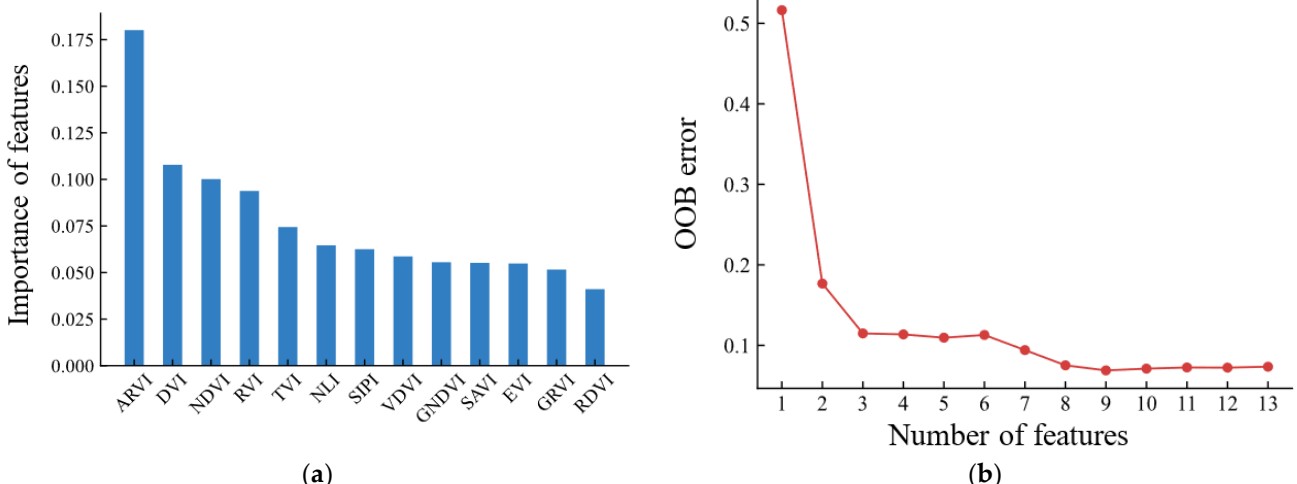

(**a**)   (**b**)

**Figure 5.** Importance ranking (**a**) and relationship with OOB error (**b**) of the vegetation indexes.

### 4.3. Textural Difference Analysis

The gray level of a GF-1 PMS image for calculating the GLCM, the direction and window size for textural features calculation, and the combination of optimized textural features are vital for the classification [33,35,44]. Therefore, this study aims to explore the optimized textural parameters, including the quantization of the gray level, textural directions, window size, and feature selection for the corn lodged area classification.

#### 4.3.1. Quantization of Gray Level

The gray level of a GF-1 PMS image determines the size of the GLCM, and a GLCM with the size of 256 × 256 will be produced from one band only with 256 gray levels. Therefore, there will be a huge computational and time-consuming effort using the original 256 gray levels to calculate the GLCM. So, the optimization of compressing grayscale is performed first. Figure 6a–c depicts the corn lodged area with the gray level of 8, 16, and 32 in the GF-1 PMS image, respectively. The boxed area in Figure 6 is the moderately lodged area, which has been confused with a severely lodged area (bright white area) in an image with eight gray levels. This confusion problem is relieved when the gray level increases to 16 gray levels, after which these three kinds of lodged areas can be distinguished clearly.

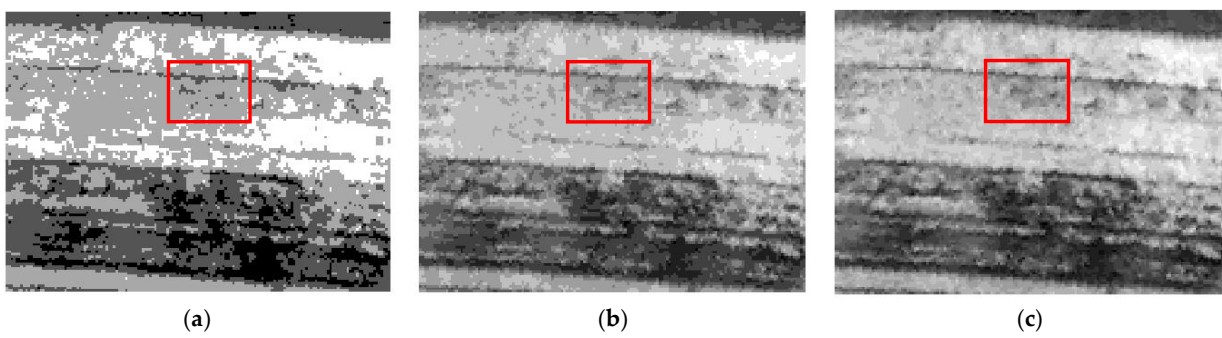

      (**a**)              (**b**)           (**c**)

**Figure 6.** Comparison of gray level with 8 (**a**), 16 (**b**), and 32 (**c**) levels in GF-1 PMS images.

With increasing the image gray level to 32 levels, the distinction between the non-lodged, moderately, and severely lodged areas is almost similar. Figure 7 shows the statistical variation of gray values in the non-lodged area, moderately lodged area, and severely lodged area with the gray levels of 8 (a), 16 (b), and 32 (c). We can see that there is obvious spectral confusion with a gray level of 8 (Figure 7a), and this problem is relieved with a gray level of 16 (b) and 32 (c). There is almost no difference in boxplots shown in (b) and (c). Based on the results of Figures 6 and 7, we use the gray level of 16 for the GLCM textural features calculation.

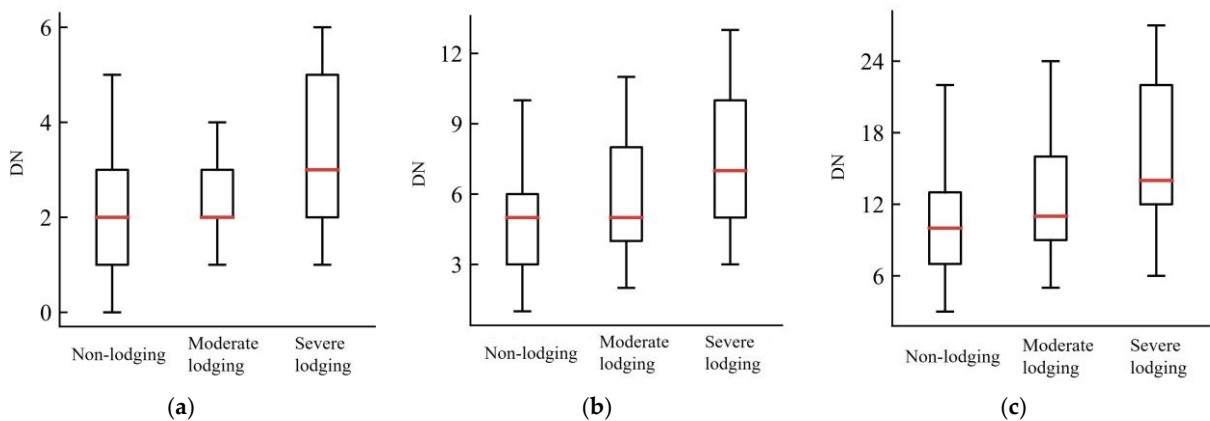

      (**a**)              (**b**)           (**c**)

**Figure 7.** Boxplot of the pixels' gray value with the gray level of 8 (**a**), 16 (**b**), and 32 (**c**).

### 4.3.2. Analysis of Textural Directions

There are varied directions for cropland, including east to west (Figure 8a), south to north (Figure 8b), northeast to southwest, northwest to southeast, and other directions. There are four kinds of directions developed in the GLCM algorithm, including 0°, 45°, 90°, and 135°. For analyzing the difference in the textural direction for the textural features calculation, the optimized eight popular textural features are all calculated in four directions, including 0°, 45°, 90°, and 135°. The lined plots and dotted plots in Figure 9a–h are the calculated statistical eight textural features in four directions for the corn planted area with the direction of east to west and south to north. Figure 9c–e shows that (1) there is an obvious textural difference in the different directions of the textural features, including correlation (COR), contrast (CON), and dissimilarity (DIS); (2) the COR value is the highest when the direction of calculating the COR feature is the same as the cropland direction; (3) the values of CON and DIS are the smallest when the directions of calculating the CON and DIS features are the same as the cropland direction. In addition to these three textural features, there is no obvious difference within the different directions. The results of the statistical analysis show varied directions in the cropland of the study area. To cover all the textural characteristics in the study area, we calculated the mean values for all textural features in four directions for classifying the corn lodged area.

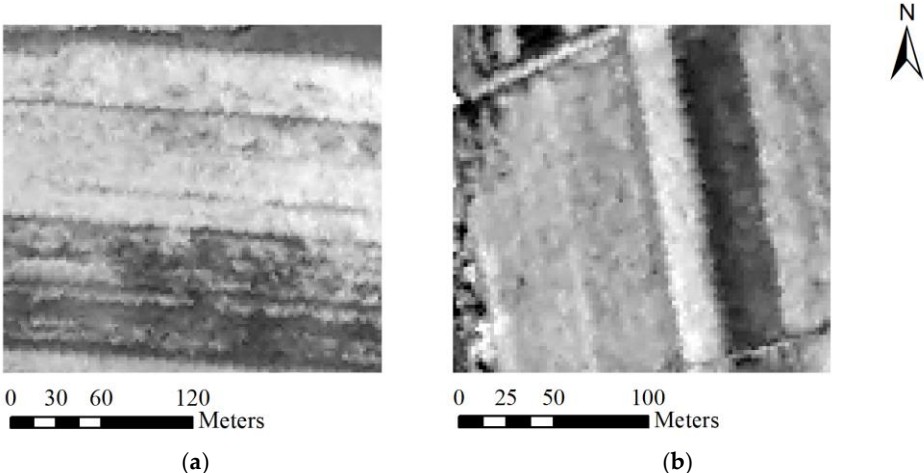

**Figure 8.** The cropland with rowing from east to west (**a**) and south to north (**b**).

### 4.3.3. Window Size for Texture Computation

Window size is essential for texture features, and windows with different sizes can capture textural information on different scales. A small window cannot cover the complete textural features of the cropland, and a too-large window will create a spectral and textural mixture. So, the window size should be determined. The window size is expressed as $(2N + 1) \times (2N + 1)$, where $N = 1\sim25$ in this study. Therefore, there are 26 window sizes used for calculating the textural features. Figure 10 shows the calculated Mean (a), VAR (b), COR (c), CON (d), DIS (e), HOM (e), ASM (f), and ENT (g) using window sizes from $3 \times 3$ to $51 \times 51$, respectively. Figure 10 shows a noticeable textural difference with the window size of $3 \times 3$ for the non-lodged area, moderately lodged area, and severely lodged area in addition to the textural features of VAR (b) and COR (c).

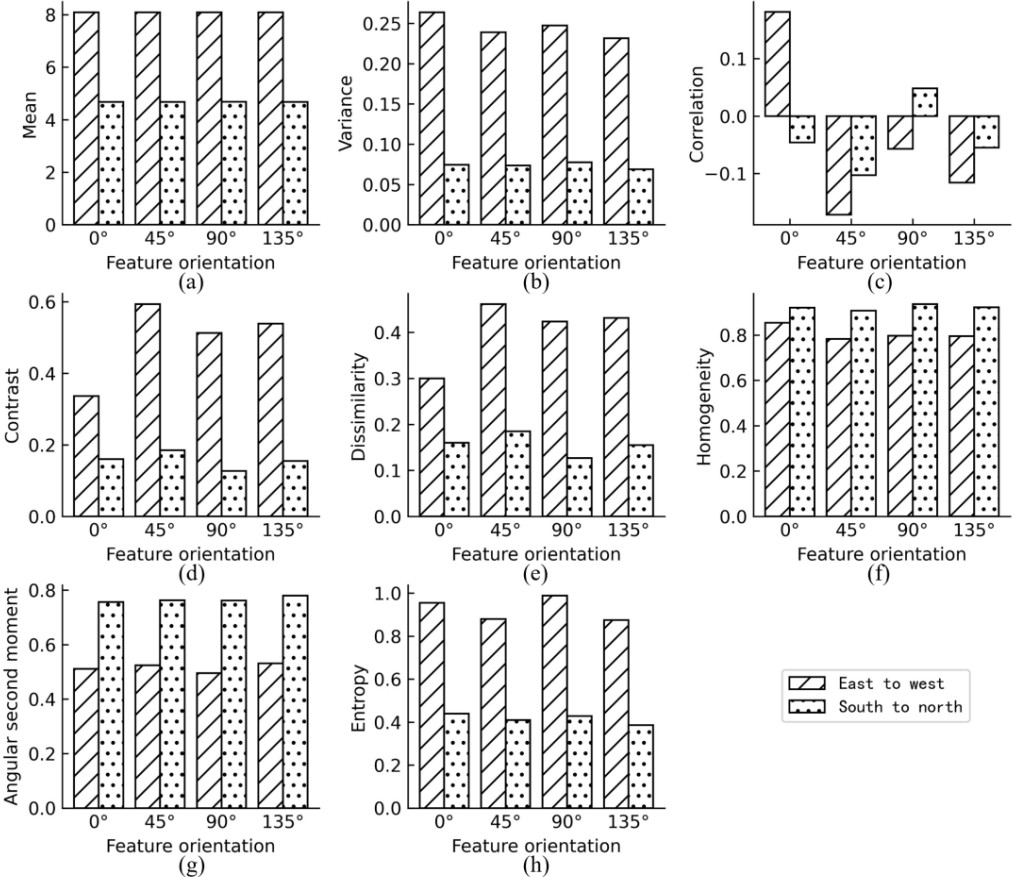

**Figure 9.** Textural features from the cropland with rowing from east to west and south to north. And (**a**–**h**) represent the textural features of mean, variance, correlation, contrast, dissimilarity, homogeneity, angular second moment, and entropy, respectively.

For quantitating the differences in the eight textural features with different window sizes, the J.M. distance introduced in Section 3.1.3 is calculated. The valid J.M. distance value lies in the range of [0.0, 2.0]. There is good separability between the two analyzed classes when the J.M. distance lies in the range of [1.9, 2.0]. A J.M. distance within [1.0, 1.9] means moderate separability, and there will be many pixels that should be classified wrongly. A J.M. distance value within [0.0, 1.0] means poor separability. Figure 11 shows the J.M. distance of the non-lodged, moderately lodged, and severely lodged areas for the textural features using window sizes from 3 × 3 to 51 × 51, where G1 is the J.M. distance between the non-lodged and moderately lodged area, G2 is that between the non-lodged and severely lodged area, and G3 is that between the moderately and severely lodged area. Figure 11 shows that the J.M. distance of G2 and G3 with a window size of 3 × 3 is bigger than 1.9 with good separability. When the window size increases, the J.M. distance of G2 and G3 decreases. So, the window size of 3 × 3 is determined as the optimal size for corn lodging identification using GF-1 PMS images. Comparably speaking, the separability of G2 and G3 is better than that of G1, which means that there is more textural mixture between the non-lodged and moderately lodged areas.

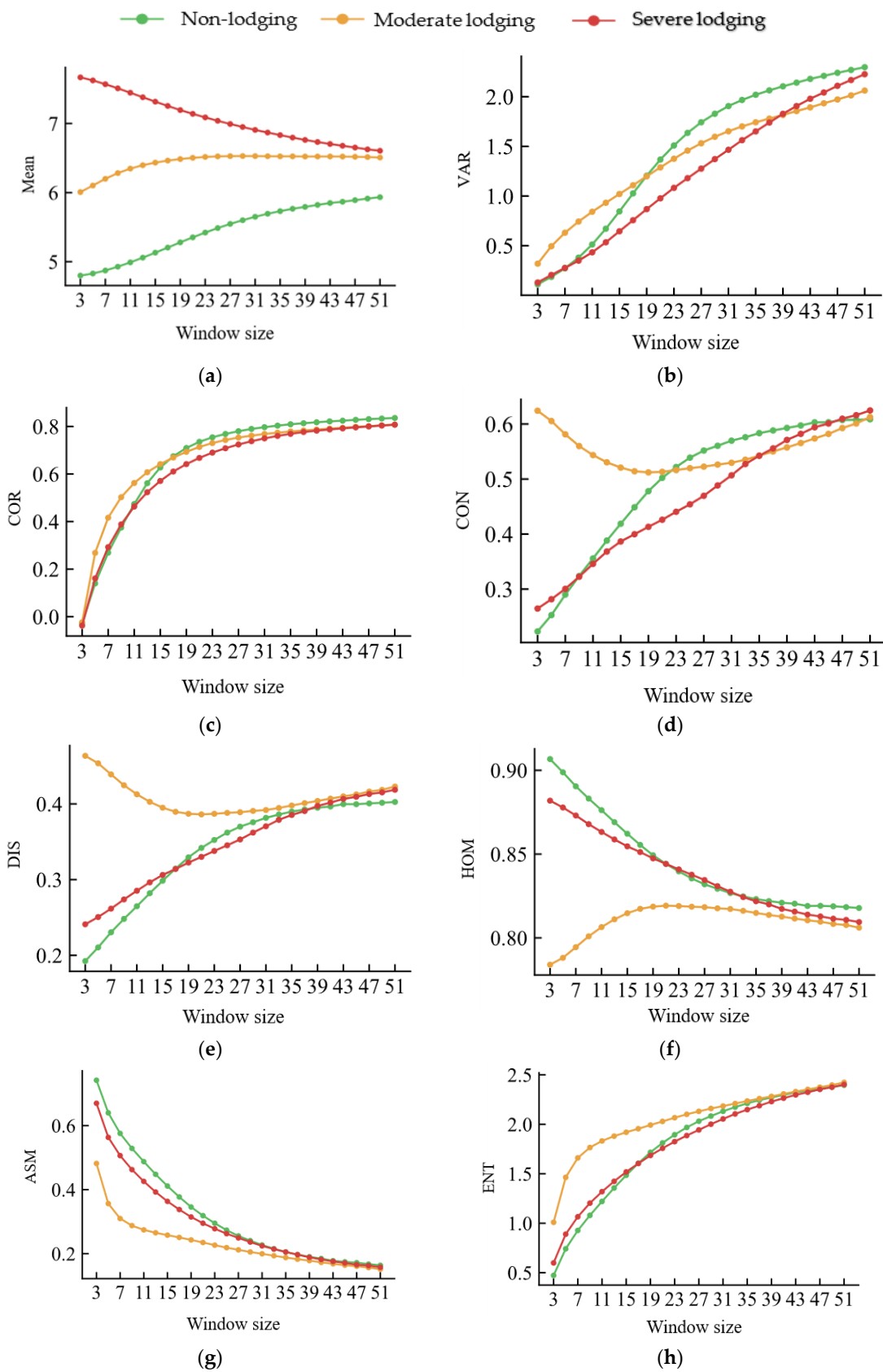

**Figure 10.** The textural difference with window sizes from 3 × 3 to 51 × 51 for the non-lodged area, moderately lodged area, and severely lodged area. And (**a**–**h**) represent the textural features of mean, variance, correlation, contrast, dissimilarity, homogeneity, angular second moment, and entropy, respectively.

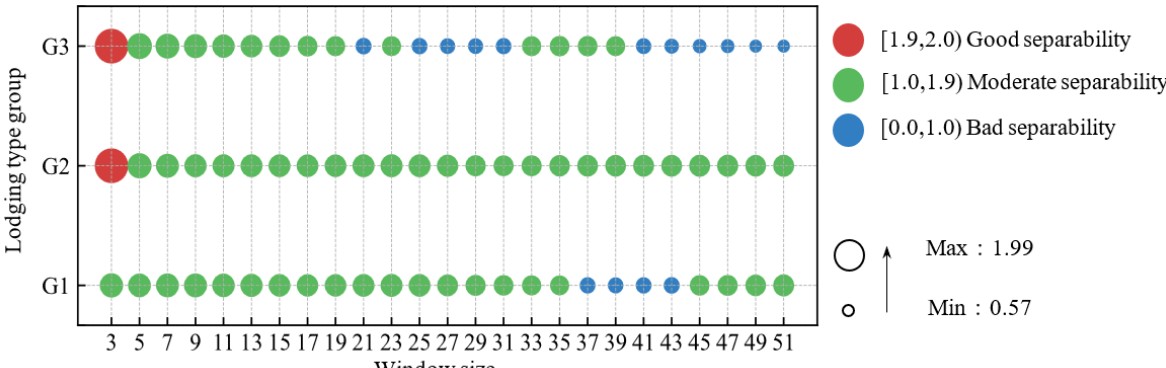

**Figure 11.** J.M. distance of the non-lodged, moderately lodged, and severely lodged areas for the textural features using window sizes from 3 × 3 to 51 × 51.

4.3.4. Selection of Textural Features

Considering the redundancy and the consumption of time and computing resources, we use the Gini index introduced in Section 3.1.2 to optimize the textural features. The importance of the eight textural features is ranked as shown in Figure 12a, which reveals that the Mean feature has the highest importance, and the importance of other textural features is ranked as COR, VAR, CON, ENT, ASM, DIS, and HOM successively. In line with the ranking of importance, one more textural feature is joined one by one for the corn lodging identification using a random forest classifier. The OOB error of the corn lodging identification is as shown in Figure 12b, which reveals that the classification error decreases to the smallest value of 0.3918 when the number of textural features reaches five. Therefore, five textural features are joined for the corn lodging identification, including Mean, COR, VAR, CON, and ENT.

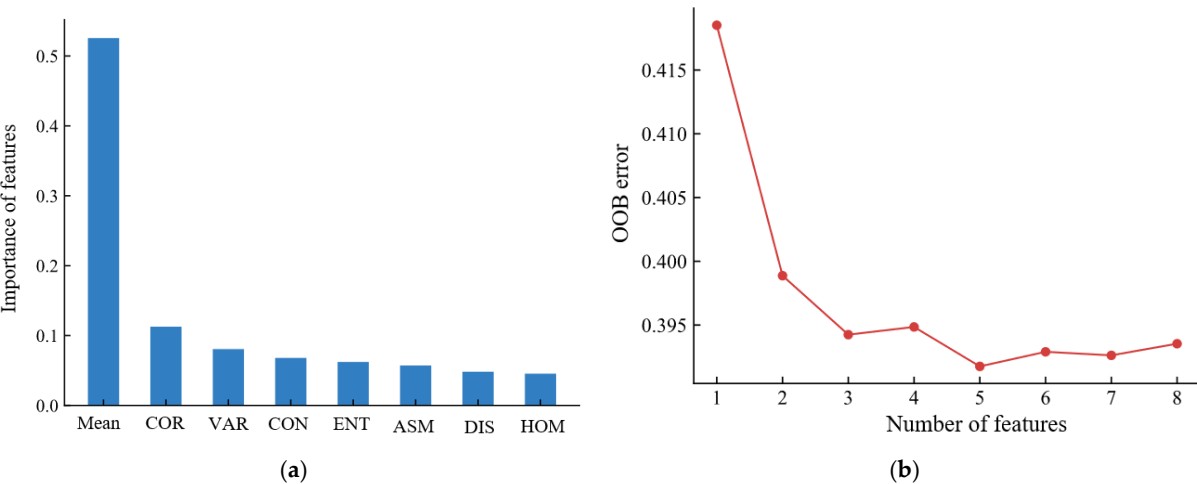

**Figure 12.** The importance ranking (**a**) and OOB error (**b**) of eight textural features.

From these four experiments, we determined that the optimized textural features for corn lodging identification using GF-1 PMS images are calculated with the gray level of 16, the average textural features using the directions of 0°, 45°, 90° and 135°, a window size of 3 × 3, and the combination of textural features including Mean, COR, VAR, CON, and ENT.

*4.4. Optimization of Machine Learning Methods*

Based on the optimized combination of the spectral bands, vegetation indexes, and textural features introduced in Sections 4.1–4.3, the optimization of five machine-learning

classifiers is performed here. As mentioned in Section 3.2, the Naive Bayes classifier is easy, and no super parameters should be determined. The number of nodes in the hidden layer of BP neural networks has been optimized. Therefore, the classifier of SVM, the random forest, and XGBoost are optimized here.

Two essential SVM parameters should be optimized: the penalty parameter $C$ and kernel function [38]. Figure 13a shows the overall accuracy using different kernel functions and $C$ values. For the kernel function, the overall accuracy value using the sigmoid kernel function is lower than that of the poly and rbf. For the $C$ value, there is almost no change in overall accuracy using the poly kernel function when the $C$ value changes, which means the $C$ value is insensitive to the poly kernel function. Therefore, the rbf kernel function is used for the corn lodging identification. Based on Figure 13a, we determined that the rbf kernel function and a $C$ value of $10^4$ are the optimized parameters of SVM. In addition, we optimize the gamma parameter of the rbf kernel function, and the gamma parameters to be selected are 0.1, 1, 3, 5, 7, and 9 respectively. The best gamma parameter compared by cross-validation is 7.

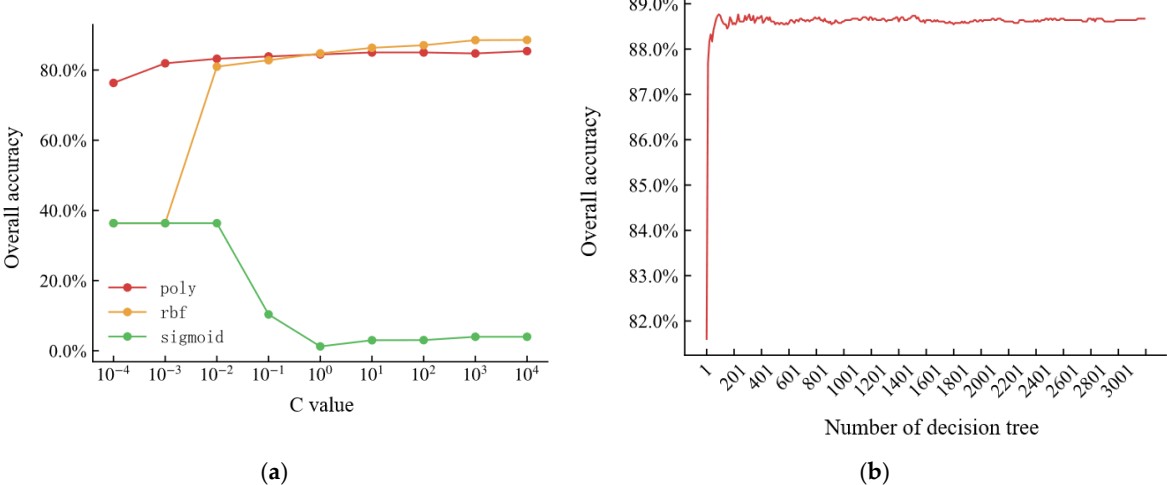

**Figure 13.** Comparison of efficiency using different kernel functions of SVM (**a**) and a different number of trees in random forest (**b**).

The number of decision trees is vital for the random forest classifier. Figure 13b shows the change in accuracy when changing the number of the random forest classifier. The overall accuracy is only 81.80% when the number of decision trees is one. The overall accuracy increases to 87.41% when the number of decision trees reaches 11. Then the overall accuracy is stabilized after the number of decision trees reaches 351, and the overall accuracy reaches a maximum of 88.92%. Therefore, the number of decision trees is configured as 351 for the corn lodging identification in this study.

In order to obtain the optimal parameters of XGboost, we used a grid search algorithm based on 10-fold cross-validation to compare the model accuracy of different parameter combinations. Then, we compare the accuracy of different parameter combinations to obtain the optimal parameter combination. The number of gradient boosted trees is 150 and the maximum tree depth is 7.

### 4.5. Classification Results of Non-Lodged, Moderately Lodged, and Severely Lodged Areas

Using the optimized image features and machine-learning classifiers, we classify the study area's non-lodged, moderately, and severely lodged areas. Tables 7 and 8 show the quantitative accuracy assessment results, including the OA, Kappa, and F1 scores for the corn lodging identification. Table 7 shows that OA and Kappa are all improved when joining the vegetation indexes to four spectral bands for the SVM, RF, BP, and XGBoost classifiers. The OA and Kappa are improved further when the textural features are joined to the four spectral bands for all four classifiers, including SVM, RF, NB, BP, and XGBoost. The

highest OA and Kappa values are reached for the combinations of spectral bands, vegetation indexes, and textural features using the SVM, RF, BP, and XGBoost classifiers. Comparably speaking, the OA and Kappa by NB classifiers are the lowest for all combinations. The lower accuracy of the NB classifier may result from the hypothesis that the NB classifier assumes that input features are independent of each other. Unfortunately, the vegetation index is computed from spectral bands, which are highly correlated. Therefore, the NB classifier is not suitable when using remote sensing images to identify corn lodging.

**Table 7.** OA and Kappa coefficients using different classifiers and a combination of image features.

| Combination of Features | SVM | | RF | | NB | | BP | | XGBoost | |
|---|---|---|---|---|---|---|---|---|---|---|
| | OA | Kappa | OA | Kappa | OA | Kappa | OA | Kappa | OA | Kappa |
| Spectral | 89.23% | 0.8382 | 89.24% | 0.8384 | 0.7549 | 0.6935 | 83.64% | 0.7549 | 88.59% | 0.8283 |
| Spectral + vegetation index | 91.09% | 0.8663 | 90.86% | 0.8628 | 0.7858 | 0.6438 | 85.7% | 0.7858 | 91.23% | 0.8685 |
| Spectral + texture | 91.16% | 0.8674 | 91.97% | 0.8792 | 0.8085 | **0.7285** | 87.25% | 0.8085 | 92.33% | 0.8850 |
| Spectral + vegetation index + texture | **93.23%** | **0.8991** | **93.81%** | **0.9069** | 0.8383 | 0.6890 | **89.24%** | 0.8383 | **93.37%** | **0.9005** |

**Table 8.** F1 scores using different classifiers and a combination of image features.

| Classifier | Combination of Features | Non-Lodged Area | Moderately Lodged Area | Severely Lodged Area |
|---|---|---|---|---|
| SVM | Spectral | 0.9167 | 0.8547 | 0.9097 |
| | Spectral + vegetation index | 0.9313 | 0.8831 | 0.9208 |
| | Spectral + texture | 0.9208 | 0.8822 | 0.9326 |
| | Spectral + vegetation index + texture | **0.9358** | **0.9107** | **0.9532** |
| RF | Spectral | 0.9152 | 0.8539 | 0.9130 |
| | Spectral + vegetation index | 0.9370 | 0.8800 | 0.9122↓ |
| | Spectral + texture | 0.9285 | 0.8936 | 0.9412 |
| | Spectral + vegetation index + texture | **0.9503** | **0.9182** | **0.9492** |
| NB | Spectral | **0.8394** | 0.6961 | 0.8503 |
| | Spectral + vegetation index | 0.8367↓ | 0.6469↓ | 0.8204↓ |
| | Spectral + texture | 0.8273↓ | **0.7369** | **0.8972** |
| | Spectral + vegetation index + texture | 0.8186↓ | 0.6923↓ | 0.8691 |
| BP | Spectral | 0.8787 | 0.7658 | 0.8682 |
| | Spectral + vegetation index | 0.8876 | 0.7966 | 0.8904 |
| | Spectral + texture | 0.8809 | 0.8265 | **0.9198** |
| | Spectral + vegetation index + texture | **0.9101** | **0.8557** | 0.9164 |
| XGBoost | Spectral | 0.9079 | 0.8433 | 0.9095 |
| | Spectral + vegetation index | 0.9468 | 0.8819 | 0.9115 |
| | Spectral + texture | 0.9388 | 0.8929 | 0.9404 |
| | Spectral + vegetation index + texture | **0.9485** | **0.9052** | **0.9490** |

Note: The symbol '↓' indicates that when using the same classification method, the classification accuracy of this feature combination is lower than that of only spectral feature combination.

Table 8 shows that the F1 scores using the RF classifier are the highest for all non-lodged and moderately lodged areas. In the severely lodged area, the F1 score of SVM is the highest. Based on the results in Tables 7 and 8, the RF classifier performance is the best. For the feature combination, all the combinations improve the F1 scores using the BP classifier. Secondly, the combinations of spectral + texture and spectral + vegetation index + texture improve the F1 scores using RF classifier with a slight decrease using the combination of spectral + vegetation index. For the NB classifier, the joining of the textural features and vegetation index decreased the F1 scores for the classification of the non-lodged, moderately lodged, and severely lodged areas.

The mapping of corn lodging in the study area using GF-1 PMS images is performed using the feature combination of spectral + vegetation index + texture and the RF classifier. Figure 14 shows the masked result of the non-lodged, moderately, and severely lodged area mapping. The corn plants in the middle of Lishu County are tall and dense, and there are more severely lodged areas in the middle of Lishu County. Conversely, the corn plants in the west of Lishu County are short and sparse, so most areas are subsequently classified as non-lodged.

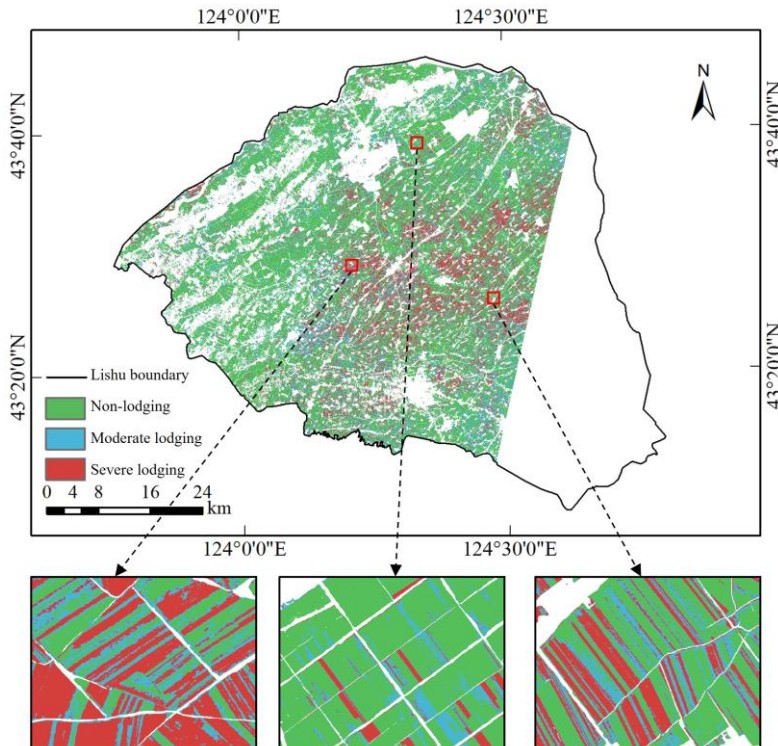

**Figure 14.** Mapping of corn non-lodged, moderately lodged, and severely lodged areas in the study area using GF-1 PMS images.

## 5. Discussion

There are challenges in identifying crop lodging using remote sensing images due to the spatial heterogeneity of the cropland and crop growth [15]. We explore the potential of Chinese GF-1 PMS images for identifying corn lodging using optimized features and optimized machine learning methods in this study. Due to lodging incidence and the coverage ability of GF-1 PMS images, there is no image collected in the southeast of Lishu County, although there are six covered and collected after lodging. Therefore, there is no identified result within the southeast of our study area in Figure 14. Our findings can cover the whole of Lishu County if there are cloudless GF-1 PMS images in the white null area. In addition, we will perform comparative experiments and validations when we collect the matched cloudless GF-1 PMS image where the corn lodging occurred.

Image features optimization and machine learning methods optimization are methods to finely depict the spatial heterogeneity of corn lodging in high-spatial-resolution GF-1 PMS images. For the textural features optimization, the moving step size when calculating the GLCM is vital in addition to the gray level, textural direction, window size, and textural features analyzed in Section 4.3. Step size refers to the interval between the base window and the moving window. Considering most lodged patches are small, crowded together, and scattered, the step size for calculating textural features is set at 1. Most of the corn lodged area is less than 5 m$^2$ in the study area, which is the size of about three pixels. In addition, the optimized 3 × 3 window size with a step size bigger than 1 will miss many

details of the lodged area. So, we use the $3 \times 3$ window size with the step size of 1 for identifying corn lodged areas in this study.

Machine learning algorithms rely on a set of parameters to construct models and make predictions. The choice of these parameters can greatly impact the accuracy of the model, and finding the optimal set of parameters is essential for achieving the best performance. In this study, we focused on the high-precision identification of corn-lodged regions and optimized the hyperparameters of the SVM, RF, and XGBoost models. For example, in the case of SVM, in addition to the main parameters of the penalty factor C and kernel function, the gamma parameter of the RBF kernel function also has a significant impact on the model's performance. In this study, we specifically adjusted the gamma parameter of the kernel function to obtain the best model. Our results showed that the optimal value of gamma can improve the overall classification accuracy of SVM by approximately 1% compared to the default value. To find the optimal parameter combination, we used a cross-validation-based grid search method, which is a commonly used technique for hyperparameter optimization. This method allows us to quickly explore different combinations of parameter values and select the one that provides the best results. However, it should be noted that as the number of parameters to adjust increases, this method becomes less efficient. In future research, alternative techniques such as random search may be considered to improve the efficiency of parameter adjustment.

## 6. Conclusions

Identifying corn lodging using Chinese GF-1 PMS images is vital for finding an efficient and fast way to classify a corn lodged area and for exploring the potential of Chinese GF series satellite images in crop growth monitoring. This study aims to find an automatic and simple machine learning method to explore how to classify stagnation areas in maize by optimizing the image features of high-spatial-resolution multispectral images of GF-1 PMS. This study evaluates and screens the importance of a large number of image features such as vegetation index and texture, and tries different feature combinations of the spectral band, spectral band + vegetation index, spectral band + texture feature, and spectral band + vegetation index + texture feature. Five machine learning algorithms including support vector machine, random forest, Bayesian, backpropagation network, and XGBoost were used to identify lodging areas of corn in combination with different feature combinations. The results of this study are as follows:

(1) The optimized textural features for corn lodging identification using GF-1 PMS images are calculated with the gray level of 16, the average textural features using the direction of $0°$, $45°$, $90°$, and $135°$, a window size of $3 \times 3$, and the combination of textural features including Mean, COR, VAR, CON, and ENT.

(2) The combination of spectral bands, optimized vegetation index, and textural features can improve the classification accuracy of high-spatial-resolution GF-1 PMS images for corn non-lodging, moderate lodging, and severe lodging areas compared with the other three feature combinations.

(3) Compared with the other four classifiers, random forest has excellent performance. It is efficient, robust, and easily identified corn non-lodging, moderate lodging, and severe lodging areas.

**Author Contributions:** Conceptualization, W.S.; funding acquisition, W.S.; methodology, X.H. and X.W.; formal analysis, X.H.; investigation, F.X. and Y.D.; data curation, F.X., Y.D., W.S. and J.L.; software, X.H. and X.W.; resources, X.H., F.X., Y.D., J.H., X.L., Y.Z. and S.M.; supervision, W.S.; validation, X.W.; writing—original draft, X.H. and W.S.; writing—review and editing, J.H., X.L., Y.Z., S.M. and J.L. All authors have read and agreed to the published version of the manuscript.

**Funding:** This research was funded by the National Key R&D Program Project (No. 2022YFD2001103), the National Natural Science Foundation of China under the project (No. 42171331), and the 2115 Talent Development Program of China Agricultural University.

**Data Availability Statement:** Not applicable.

**Acknowledgments:** We thank the journal's editors and anonymous reviewers for their kind comments and valuable suggestions to improve the quality of this paper.

**Conflicts of Interest:** The authors declare no conflict of interest.

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
