# Peer review of "Identifying Corn Lodging in the Mature Period Using Chinese GF-1 PMS Images"

_remotesensing, doi:10.3390/rs15040894_

Round 1

Reviewer 1 Report

This study aims to explore the potential of Chinese GF-1 PMS high spatial resolution image for corn lodging monitoring and find a robust and efficient way to accurately and quickly understand corn lodging. The random forest is an efficient, robust, and easy classifier to understand the corn lodging with the F1-score of 0.95, 0.92, and 0.95 for non-lodged, moderate lodged, and serious lodged areas. 

P244, In the Figure3 the model optimization should be using the training samples, then for classifying and mapping with the testing samples.

 P263 When using the random forest to classification, the importance and OOB error would be calculated. This paper shows that “vegetation index would combine the image features one by one”. But during the processing of training dataset, this part is missed and should be added.

 P389 During using the SVM, the C and gamma both are the same important, if you would use the ‘rbf’ kernel function. You should optimize the gamma value. And the random forest is an old method. The new boosting algorithm should be used , such as xgboost, because it is often used in practical applications.

 P414 I'm skeptical of the results of this model. Since your features are extracted using random forest, SVM and BP modeling are conducted on the basis of the extracted features, and the model results are inevitably poor compared with RF. That is to say, using RF to screen features for SVM and BP modeling has problems for modeling fairness, and the model results are not comparable.

Experiments must be supplemented to include all features, as well as other feature extraction methods for comparative analysis.Only using RF as feature extraction, the content is too simple and no scientific.

 P437-438 “methods” should be changed “method”

 The conclusion should be summarized the better.  

Author Response

Dear Reviewer,

We thank you very much for reviewing our paper and giving your comments. We have revised the contents according to your suggestions. The point-by-point response to the comments is listed in the supplement.

Looking forward to hearing from you.

Best regards,

Wei Su

Reviewer 2 Report

The study aims to explore the potential of Chinese high-resolution GF-1 PMS imagery for monitoring corn lodging. The work uses modern methods such as machine learning, support vector machine, random forest, naive Bayes method. The results of the study show that the combination of spectral bands, optimized vegetation indices and texture characteristics classifies corn lodging with an overall accuracy of 93.81%.

At the same time, I want to note that the result was obtained for a specific case (for one date). The result obtained does not mean that the selected vegetation indices (line 275), textural features (line 369) are optimal at other times or in another survey year. More research is needed to verify this.

Specific comments:

The paper does not have a "Discussion" section. Why?

Line 257. Figure 4. On the scales, the font is of different sizes, the text is superimposed on the values of the “Spectral reflectance” scale.

 To confirm the results of the experiment (September 26), it is necessary to conduct additional data analysis on other dates.

 How many times was RF repeated to calculate mean OA?

Author Response

(The authors gave the same response as above.)

Reviewer 3 Report

This Manuscript presents an interesting and well elaborated research about the identification and mapping of corn lodged areas by using images from the Gaofen-1 satellite. However, extensive editing of English language is suggested. The brief summary follows:

- It should be used “moderately” and “severely lodged” instead of “moderate lodged” and “serious lodged”.

- Please check if it is GF-1 PMS or GF-1 PMC. Please provide some reference.

- In lines 66-67, I suppose that the correct word should be ‘hit” instead of “heat”.

- Try to avoid starting sentences with AND.

- Sentence, line 122-124 is not clear. Please consider rephrasing.

- Try to avoid phrases like “as we all know”.

- Instead of “testing” samples better use “validation”.

- Please consider rephrasing “understand corn-lodged areas”. Instead use “identify” or “classify”, or similar more appropriate words.

- Sentence, line 156-157 is not clear. Please consider rephrasing.

- In line 163, consider using “major importance” instead of “big importance”.

-  Sentence, line 168-169 is not clear. Please consider rephrasing.

-  Sentence, line 178-180 is not clear. Please consider rephrasing.

-  Sentence, line 190-192 is not clear. Please consider rephrasing. It is not clear are those methods used in your study or in general.

-  Sentence, line 196-198 is not clear. Please consider rephrasing.

- Sentence, line 231-234 is not clear. Please consider rephrasing. Please use something like “are presented in Table 6” or similar.

-  Sentence, line 261-263 is not clear. Please consider rephrasing.

- In sentence, line 318-319, rephrase “Our statistical results”. Use something like “The results of the statistical analysis” or similar.

-  Sentence, line 388-389 is not clear. Please consider rephrasing.

-  Sentence, line 412-413 is not clear. Please consider rephrasing.

I recommend this paper to be published after major revision.

Author Response

(The authors gave the same response as above.)

Round 2

Reviewer 1 Report

This manuscript has been revised, and the overall modification is satisfactory.

I agreed to accept.

Reviewer 2 Report

Dear Authors

Thank you very much for answering all my queries, i do not have any further queries regarding the manuscript.

Reviewer 3 Report

Thank you for updating your manuscript. I have no further comments.